# Sex Specific Alterations in α4*Nicotinic Receptor Expression in the Nucleus Accumbens

**DOI:** 10.3390/brainsci8040070

**Published:** 2018-04-19

**Authors:** Joan Y. Holgate, Josephine R. Tarren, Selena E. Bartlett

**Affiliations:** Institute of Health and Medical Innovation, Queensland University of Technology, Translational Research Institute, 37 Kent St, Woolloongabba, QLD 4102, Australia; Joan.holgate@anu.edu.au (J.Y.H.); josephine.tarren@hdr.qut.edu.au (J.R.T.)

**Keywords:** nicotinic receptors, nucleus accumbens, stress, varenicline, maternal separation

## Abstract

*Background*: The mechanisms leading from traumatic stress to social, emotional and cognitive impairment and the development of mental illnesses are still undetermined and consequently there remains a critical need to develop therapies for preventing the adverse consequences of traumatic stress. Research indicates nicotinic acetylcholine receptors containing α4 subunits (α4*nAChRs) are both impacted by stress and capable of modulating the stress response. In this study, we investigated whether varenicline, a partial α4β2*nAChR agonist which reduces nicotine, alcohol and sucrose consumption, can reduce stress, a driving factor in substance use disorders. We also examined the effect of stress on nucleus accumbens (NAc) α4*nAChR expression. *Methods*: Transgenic mice with fluorescent tags attached to α4*nAChRs were administered varenicline and/or yohimbine (a pharmacological stressor) and plasma corticosterone and NAc α4*nAChR expression were measured. A separated group of mice were exposed to maternal separation (MS) during post-natal day (P) 2–14, then restraint stressed (30 min) at six weeks of age. Body weight, anxiety-like behaviours (elevated plus maze), plasma corticosterone and NAc α4*nAChR levels were measured. *Results*: Varenicline attenuated yohimbine-induced plasma corticosterone increases with no effect on NAc α4*nAChR expression. MS reduced unrestrained plasma corticosterone levels in both sexes. In females, MS increased body weight and NAc α4*nAChR expression, whereas, in males, MS and restraint caused a greater change in anxiety-like behaviours and plasma corticosterone levels. Restraint altered NAc α4*nAChR expression in both male and female MS mice. *Conclusions*: The effects of stress on NAc α4*nAChR are sex-dependent. While varenicline attenuated acute stress-induced rises in corticosterone levels, future studies are required to determine whether varenicline is effective for relieving the effects of stress.

## 1. Introduction

It is inevitable; at some point in our lives, we will experience stress. From conception to death, we are exposed to environmental stimuli that we may perceive as stressful. While the acute stress response can help us avoid and respond to dangerous situations, chronic exposure to stress and its associated excessive glucocorticoids release can have long-term negative health consequences (for review, see [1]). Particularly, for children, exposure to traumatic stress during development can negatively impact their educational and socioeconomic outcomes, leading to a lifetime of poor health and a dramatically shortened life span (for review, see [2]). In fact, children exposed to multiple adverse early life events (e.g., abuse and neglect) are 4–12 times more likely to develop alcohol use disorders (AUDs), substance use disorders, depression and attempt suicide; 2–4 time more likely to smoke, have poor self-rated health, have more than 50 sexual partners and a sexually transmitted disease; and are approximately 1.5 times more likely to be physically inactive and severely obese [3]. As such, there is a critical need to improve our understanding of how stress impacts the brain during childhood and develop novel interventions and therapies for preventing its lifelong adverse health consequences. 

While there are significant scientific gaps in our understanding of the mechanisms leading from early life stress (ELS) to social, emotional and cognitive impairment and the development of addiction and other mental illnesses (e.g., post-traumatic stress disorder, schizophrenia, substance use disorders and depression), research indicates that stress-induced changes to the structure and function of the nucleus accumbens (NAc) may be the cause of symptom development [4,5,6,7,8,9,10,11,12]. A common characteristic of many ELS-associated mental disorders is a dysfunction or loss of dopaminergic signalling which leads to altered cholinergic signalling within the NAc or vice versa (for reviews see [13,14,15]. For example, in schizophrenia, disruptions to dopaminergic signalling alters cholinergic transmission between the NAc and prefrontal cortex (PFC) (for review, see [13]). In AUDs, alcohol triggers the release of dopamine in the NAc via cholinergic neurons, and changes in dopamine levels in the NAc lead to cravings, escalations in alcohol consumption and relapse to harmful consumption behaviours (for reviews see [16,17]). However, the exact mechanism by which glucocorticoids modulate this process remains to be elucidated. 

Research has established that the release of dopamine can be modulated via the binding of acetylcholine (ACh) to nicotinic acetylcholine receptors (nAChRs) [18,19]. Studies also indicate that stress can modulate the expression and function of nAChRs in the brain. Exposure to prenatal stress increases expression of alpha 4beta 2 containing (α4β2*) nAChRs in the hippocampus of rats [20] and the application of corticosterone alters ACh-dependant currents in PC2 cells [21]. We also know that nAChRs can control the release of glucocorticoids. Administration of TC-2559, an α4β2*nAChR partial agonist, increases urinary corticosterone levels [22]. Research by Yamanashi and colleagues indicated that nAChRs in the NAc are impacted in stress. They showed that mecamylamine, a non-selective nAChR antagonist, could block foot-shock stress-induced dopamine release in the NAc [19]. However, it is undetermined which nAChR subtypes are involved.

In this study, we investigated the role of α4*nAChRs in stress. First, we explored the involvement of α4*nAChRs in stress. We administered varenicline, a partial α4β2*nAChR agonist, prior to yohimbine (a pharmacological stressor) and measured yohimbine-induced changes in plasma corticosterone levels and NAc α4*nAChR expression. We also used the two-hit model to explore the impact of ELS on NAc α4*nAChR expression. Transgenic mice (with a yellow fluorescent protein tagged to α4 subunits of nAChRs) were exposed to ELS using the maternal separation model (post-natal days (P) 2–14) then, at six weeks of age, 30 min of restraint stress was applied and changes in NAc α4*nAChR expression were measured.

## 2. Materials and Methods

### 2.1. Animals and Housing

All mice were housed in climate-controlled rooms on a 12 h light cycle (lights on at 7:00 a.m. and off at 7:00 p.m.) with food and water available ad libitum. The transgenic α4YFP mice (α4 nAChR subunit tagged with yellow fluorescent protein (YFP)), generated by the Lester Laboratory (Caltech, Pasadena, CA, USA), had been backcrossed on a C57BL/6J background [23]. Receptor function was maintained following the insertion of fluorescent proteins into the intracellular M3-M4 intracellular loop of the α4 subunit. The tagged α4 nAChRs display similar localization patterns in the brain and are under the control of the same promoters, enhancers and trafficking mechanisms as the wild type α4 subunit [24]. The mice used in this study were generated from homozygous breeding pairs and have been shown to be similar to wild-type mice [24]. Mice were weaned at 21 days of age and housed (groups of 3–5) in standard cages (Tecniplast, Buguggiate, VA, Italy) with wood chip bedding material, cotton nestlets and cardboard cubby houses. Mice for breeding were housed in pairs until a plug was observed; the male was then removed. All experimental procedures (described below) occurred between 9:00 a.m. and 12:00 p.m. and all mice were allowed at least 1 h to habituate to the test room before experiments commenced. A pseudorandom Latin square design was used to control for random effects such that each batch contained at least one mouse from each test group and the order of testing was different for each batch. The experimental procedures followed the ARRIVE guidelines and were approved by the Queensland University of Technology Animal Ethics Committee (1600000698 and 1600000686) and the University of Queensland Animal Ethics Committee (TRI/QUT/122/14 and QUT/PACE/097/16/QUT), in accordance with the National Institutes of Health (NIH) guidelines for the care and use of laboratory animals.

### 2.2. Varenicline and Yohimbine Administration

Male mice (8 weeks old) were administered vehicle (saline) or 2 mg/kg (s.c.) varenicline (Sigma, Sydney, NSW, Australia) 15 min prior to receiving a vehicle or 2 mg/kg (i.p.) yohimbine (Tocris, Noble Park, VIC, Australia) injection. Approximately 100 μL of tail blood was collected into Ethylenediaminetetraacetic acid (EDTA)-treated tubes, under isoflurane anaesthesia (2–3%), 1 h following the last injection. Twenty hours later, a lethal injection of Pentobarbital was administered and once anaesthetized, a second tail blood sample was collected. Immediately following blood collection, transcardial perfusion was performed using phosphate buffered saline (PBS), the brain was harvested and the nucleus accumbens was dissected on ice.

### 2.3. Maternal Separation Procedure

On P2, the litter and mother were transported to the behavioural suite in their home cage. The mother was placed into a standard cage containing corncob bedding material and shredded paper nestlets. The litter was placed in a separate cage (with similar bedding materials) on a heat pad. After 3 h, both the mother and litter were returned to their home cage. For control litters, both the mother and litter were placed together in the same cage containing corncob bedding material and shredded paper nestlets for 3 h. This process was repeated Monday through Friday from P2 to P14 and all litters underwent this treatment 8–9 times. Litters were housed with their mother undisturbed from P15 to P21. The offspring were then weaned by sex into groups of 3–5 per cage until 5 weeks of age. At 5 weeks of age, the offspring were individually housed and given at least a week to habituate to the new housing conditions. At 6 weeks of age, restraint stress was applied and anxiety-like behaviours measured on the elevated plus maze. Handling procedures and housing conditions for control and maternal separation mice were identical, except during the ELS conditioning period (P2–P14, described above).

### 2.4. Restraint Stress

Restraint stress was applied as previously described [25]. The mice were transported to the behavioural suite in their home cage between 7:00 and 8:00 a.m. and given at least 1 h to habituate to the room and dimmed lighting conditions. Mice were placed in a restraint tube for 30 min which consisted of a 50 mL falcon tube with the base was cut off 5 mm from the end to create a nose hole. The restraint allowed the mouse to move forwards and backward but did not permit the mouse to turn head to tail. Following the 30 min of restraint, the mice were returned to their home cage for 30 min before elevated plus maze testing commenced. 

### 2.5. Elevated Plus Maze

The elevated plus maze test was performed as previously described [26]. The mice were gently picked up by the tail and placed in the centre of the maze facing an open arm. The mice could explore the maze freely for 5 min. Mice were then immediately anaesthetized with isoflurane and 100 μL of tail blood was collected (as described above). The maze (San Diego Instruments, San Diego, CA, USA) was made of white plastic and consisted of 2 open and 2 closed arms, joined by a central platform, to form a plus shape. The maze was elevated 40 cm above the floor and the arms of the maze were 30 cm long and 5 cm wide. The walls of the closed arms were 15 cm high. A camera (San Diego Instruments, San Diego, CA, USA), positioned above the maze, recorded the activity of the mice. Anymaze software (version 4.99m, Stoelting, Wood Dale, IL, USA) was used to analyse the video recordings to determine the time spent on each arm, the number of entries made to each arm and the distance travelled. The time spent on the open arm was expressed as a percentage of the total time spent on the maze. Mice were tested on the elevated plus maze approximately 20 h before euthanasia.

### 2.6. Sample Collection and Processing

After transcardial perfusion (described above), the brain was harvested and the nucleus accumbens was dissected on ice. The tissue was frozen on dry ice and stored at −80 °C. Blood samples were incubated on ice for 20 min then centrifuged (4000 rpm for 20 min at 4 °C) and the plasma collected, aliquoted and stored at −80 °C. Tissue samples were homogenized in PBS, diluted in 2x lysis buffer containing Halt protease and phosphatase inhibitor cocktail (Thermo Scientific, Waltham, MA, USA) and centrifuged at 10,000 rpm for 20 min at 4 °C. The supernatant was removed, and the total protein concentration determined using the Bradford protein assay (BioRad, Gladesville, NSW, Australia). Samples were diluted (20 μg/μL) in Laemmli sample buffer (BioRad, Gladesville, NSW, Australia) containing dl-dithioreitol (DTT, Sigma, Sydney, NSW, Australia) and incubated at 37 °C for 30 min.

### 2.7. Corticosterone Measurements

Corticosterone measurements were performed using a commercially available corticosterone ELISA kit (Cat # ADI-900-097, Enzo Life Sciences, Farmingdale, NY, USA). Plasma samples were first incubated in equal volumes with 1:100 steroid displacement reagent then assay buffer was added until a final dilution of 1:40 was achieved. Standards and diluted samples were assayed in duplicate and each wells absorbance measured at 405 nm (FLUOstar OPTIMA, BMG Labtech, Ortenberg, Germany). Corticosterone concentration was calculated from the standard curve following correction for background and nonspecific binding and conversion to percent bound. Mice with unrestrained plasma corticosterone levels above 50 ng/mL were excluded from analysis as plasma corticosterone above this level indicate there may have been a stress response to the blood collection procedure. Two female MS, one male control and 2 male MS mice were excluded based on this criterion.

### 2.8. Protein Measurements

Proteins were separated using SDS-PAGE with 4–20% tris-glycine gels (BioRad, Gladesville, NSW, Australia) and transferred under cold conditions to a PVD-F membrane (BioRad, Gladesville, NSW, Australia). Membranes were blocked in phosphate-buffered saline containing 5% milk and 0.05% Tween 20 (Sigma, Sydney, NSW, Australia) then probed with primary antibodies at 4 °C overnight. Monoclonal mouse anti green fluorescent protein (GFP, 1:500, cat # 2955, Cell Signaling, Danvers, MA, USA) and mouse monoclonal anti-GAPDH antibody (1:10,000, MA5-15738, Pierce, ThermoFisher Scientific, Waltham, MA, USA) were used to label the proteins of interest. Anti-mouse Dylight 800-conjugated secondary antibody (Cat # 610-745-002, 1:10,000, Rockland Immunochemicals, Pottstown, PA, USA) was used for protein detection with the Odyssey Infrared Imaging System (LI-COR Biosciences, Lincoln, NE, USA). Band densities were measured using Odyssey Application Software version 2.0.40 (LI-COR Biosciences, Lincoln, NE, USA) and the integrated intensity was converted to the percentage of GAPDH expression using Excel (Microsoft 2013, Albuquerque, NM, USA) software. 

### 2.9. Statistics

Statistical analysis was performed using GraphPad Prism software (version 6, La Jolla, CA, USA) or IBM SPSS Statistics (version 21, Armonk, NY, USA). Unpaired, two-tailed Student’s T-test was used to compare control and maternal separation groups for body weight for each sex. One-way ANOVA with Bonferroni’s post hoc test was used to compare the effects of varenicline and yohimbine on plasma corticosterone and NAc α4*nAChR expression. Two-way ANOVA was used to compare the effects maternal separation and sex on the percentage change in NAc α4*nAChR expression (from unrestrained to restrained levels). Three-way ANOVA was applied for all other comparisons. The two-stage step up method of Benjamini, Kregent and Yekutieli (BKY) with a false discovery rate of 0.1 was used for post hoc analysis of P values following two-way and three-way ANOVA. All results are expressed as mean ± standard error of the mean (SEM). 

## 3. Results

### 3.1. Varenicline Attenuates Yohimbine-Induced Increases in Plasma Corticosterone

Since varenicline is a partial agonist at α4β2*nAChRs and previous research indicates that modulation of α4β2*nAChRs can alter urinary corticosterone (Loomis and Gilmour, 2010), we investigated whether varenicline could attenuate yohimbine-induced increases in plasma corticosterone levels (Figure 1A). Using one-way ANOVA with Bonferroni’s post hoc test (*F*(3,40) = 9.06, *p* = 0.0001), we found that the administration of varenicline (2 mg/kg) alone had no effect on plasma corticosterone levels. Administration of yohimbine (2 mg/kg) alone caused an increase in plasma corticosterone compared to vehicle (*p* < 0.001) and varenicline alone (*p* < 0.001). When varenicline was administered prior to yohimbine, plasma corticosterone levels were attenuated compared to yohimbine alone (*p* < 0.05) and were similar to vehicle (*p* > 0.05, ns) and varenicline alone (*p* > 0.05, ns). 

Next, we assessed whether administration of varenicline and yohimbine altered α4*nAChR expression in the NAc (Figure 1B,C). Using one-way ANOVA with Bonferroni’s post hoc test, we found no effect of any of the drug combinations (*F*(3,18) = 2.15, *p* = 0.130) on α4*nAChR expression in the NAc. Although the effect was not significant, there was a trend for the administration of varenicline alone and yohimbine alone to increase NAc α4*nAChR expression and their combined treatment to attenuate this effect.

### 3.2. Maternal Separation Alters Body Weight in Female Mice Only

Previously, we have shown different effects of short and long-term exposure to ethanol and sucrose on the brain [27] and that the efficacy of varenicline (and other nAChR modulating compounds) improved following long term consumption [28,29]. Given this, it was possible that chronic stress would be necessary to alter NAc α4*nAChR expression in the brain. To model chronic exposure to stress, we chose the maternal separation model. This model exposes mice to repeated stress during early life. It is an appropriate preclinical model considering the strong link between ELS and poor mental health outcomes in human studies [3]. First, we assessed the effectiveness of maternal separation protocol by determining whether we could detect a known MS phenotype. Previous studies show maternal separation has long lasting effects on the body weight of the offspring [30,31,32,33]. We compared the body weight of our six-week-old control and MS mice using the unpaired two-tailed Student’s *T*-test (*n* = 24–26). In female mice (Table 1), exposure to MS caused a significant increase in body weight (*p* = 0.036). However, there was no effect of MS on body weight in male mice (Table 1) (*p* = 0.171, ns).

### 3.3. Restraint Stress Increases Anxiety-Like Behaviours in Female MS Mice

Next, we wanted to compare anxiety-like behaviours following MS in female and male mice. Previous publications suggest that MS alone may not be sufficient to impact anxiety-like behaviours [34,35,36,37], that the application of a second acute stressor is necessary to produce changes in anxiety-like behaviours (known as the two-hit model) [33,34]. Therefore, at six weeks of age, we measured anxiety-like behaviours on the elevated plus maze following a single 30 min restraint stress in control and MS mice. We found no interaction of sex, maternal separation (MS) and restraint (three-way ANOVA: *F*(1,64) = 0.978, *p* = 0.326), MS and restraint (*F*(1,64) = 0.242, *p* = 0.624), sex and MS (*F*(1,64) = 1.305, *p* = 0.258) and sex and restraint (*F*(1,64) = 3.212, *p* = 0.078) and no effect of restraint alone (*F*(1,64) = 0.102, *p* = 0.750), MS alone (*F*(1,64) = 2.815, *p* = 0.098) and sex alone (*F*(1,64) = 0.049, *p* = 0.825) on the percentage of time spent on the open arm (Figure 2A). We also examined the number of entries to the open arm and found no interaction of sex, MS and restraint (three-way ANOVA: *F*(1,64) < 0.0001, *p* = 0.989), MS and restraint (*F*(1,64) = 0.606, *p* = 0.439), sex and MS (*F*(1,64) = 1.315, *p* = 0.256) and no effect of MS alone (*F*(1,64) = 1.254, *p* = 0.267) and sex alone (*F*(1,64) = 1.085, *p* = 0.302) (Figure 2B). However, there was an interaction of sex and MS (*F*(1,64) = 5.468, *p* = 0.023) and an effect of restraint alone (*F*(1,64) = 9.799, *p* = 0.003). Post hoc analysis using BKY’s test showed a significant effect of restraint in female MS but not female control or male control or MS mice. Female unrestrained control and MS mice had higher plasma corticosterone levels than unrestrained male control and MS mice and male restrained MS mice. We also found no interaction of sex, maternal separation (MS) and restraint (three-way ANOVA: *F*(1,64) = 0.056, *p* = 0.814), MS and restraint (*F*(1,64) = 0,476, *p* = 0.493), sex and MS (*F*(1,64) = 1.109, *p* = 0.296) and sex and restraint (*F*(1,64) = 2.133, *p* = 0.149) and no effect of restraint alone (*F*(1,64) = 0.440, *p* = 0.123), MS alone (*F*(1,64) = 0.407, *p* = 0.526) and sex alone (*F*(1,64) = 1.239, *p* = 0.250) on the distance travelled on the maze (Figure 2C).

### 3.4. ELS Has Sex-Specific Effects on Plasma Corticosterone Levels in Response to Stress

Since we only detected small effects on anxiety-like behaviour with our modified two-hit model, we also examined plasma corticosterone levels. Analysis using three-way ANOVA showed a significant interaction of sex, MS and restraint (*F*(1,59) = 6.930, *p* = 0.011) and an effect of restraint alone (*F*(1,59) = 5.525, *p* = 0.022) and sex alone (*F*(1,59) = 39.535, *p* < 0.0001). Post hoc analysis with BKY’s test showed an increase in plasma corticosterone levels following restraint in Female control (*p* = 0.003) and MS mice only (*p* = 0.029, Figure 3A). However, there was no effect of MS in unrestrained or restrained mice of either sex. Unrestrained female control and MS mice had greater plasma corticosterone levels than unrestrained male MS mice and restrained female control and MS mice had higher plasma corticosterone levels than all four groups of male mice.

Next, we examined the magnitude of the plasma corticosterone response to restraint stress. We corrected for potential differences in basal levels between the MS and control groups by expressing the plasma corticosterone levels as the percentage change from basal levels ((test corticosterone levels minus basal corticosterone levels) divided by basal corticosterone levels times 100). There was a significant interaction of sex, MS and restraint (*F*(1,59) = 8.874, *p* = 0.004) and an effect of restraint (*F*(1,59) = 7.579, *p* = 0.008), MS (*F*(1,59) = 4.357, *p* = 0.041) and sex (*F*(1,59) = 8.274, *p* = 0.006) alone (three-way ANOVA, Figure 3B). Female unrestrained controls showed a smaller change from basal plasma corticosterone levels than all other female groups and restrained male MS mice using BKY post hoc analysis. Restrained male MS mice also displayed a greater change from basla plasma corticosterone levels than all other male groups, indicating a significant effect of restraint in male MS (*p* = 0.042) and not control mice (*p* > 0.05). 

### 3.5. MS Attenuates Restraint-Induced Decreases in NAc α4*nAChR Expression in Male Mice

After confirming we could detect phenotypic differences in both males and female mice using our modified two-hit model, we examined the impact restraint stress and MS had on NAc α4*nAChR expression. Analysis using three-way ANOVA showed a significant interaction of MS and restraint (*F*(1,52) = 5.594, *p* = 0.022). Exposure to restraint reduced NAc α4*nAChR expression in male control mice only (Figure 4A, *p* = 0.039, BKY’s post hoc test), indicating MS attenuates the effect of restraint on NAc α4*nAChR expression. To explore whether the magnitude of the effect of restraint stress on NAc α4*nAChR expression was different following exposure to MS we calculated the percentage change in expression from the average of the unrestrained expression for each group (similar to plasma corticosterone above). Two-way ANOVA showed a significant interaction of sex and MS (*F*(1,19) = 7.153, *p* = 0.015) and an effect of sex alone (*F*(1,19) = 12.500, *p* = 0.002). The percentage change in NAc α4*nAChR expression from unrestrained conditions was greatest in female control mice (BKY’s post hoc test, *p* < 0.01) for all three groups (Figure 4B). There was no effect of MS on the magnitude of change in NAc α4*nAChR expression in male mice. In female mice, the magnitude of change in expression was positive in controls and negative in MS mice. 

## 4. Discussion

We examined the role of α4*nAChRs in stress. Firstly, we explored their role in acute stress by administering yohimbine, a pharmacological stressor, following an injection of varenicline, a partial agonist at α4β2*nAChRs, and measured plasma corticosterone levels and α4*nAChR expression in the NAc. Administration of varenicline significantly attenuated yohimbine-induced increases in plasma corticosterone levels. Our findings support previous studies indicating that manipulation of α4β2*nAChRs can alter corticosterone levels. Loomis and Gilmour found that administration of nicotine (non-selective nAChR agonist) or TC-2559 (an α4β2*nAChR partial agonist) or forced swim stress increased urinary corticosterone levels [22]. Furthermore, they found that administration of mecamylamine (a non-selective nAChR antagonist) attenuated the forced swim stress-induced increase in corticosterone levels [22]. Varenicline is similar to TC-2559 in that it is a partial agonist at α4β2*nAChRs. However, unlike TC-2559, it can also act on other nAChR subtypes (particularly α3β4*nAChRs, see [38]), which could explain its ability to mimic the actions of mecamylamine on corticosterone levels in the previous study, yet produce no effect on plasma corticosterone levels when administered alone. 

Importantly, we were not able to detect a change in NAc α4*nAChR expression with any of the varenicline-yohimbine treatment combinations. Data from our laboratory suggests that long-term consumption of sucrose is required to alter the morphology of NAc neurons [27] and the longer an animal consumes ethanol or sucrose, the greater the efficacy of compounds which reduce consumption [28,29]. Given this, it is possible that chronic exposure to stress is required to alter NAc structure and function and for varenicline to produce a detectable change in α4*nAChR expression. Together, the unaltered NAc α4*nAChR expression following yohimbine only treatment and the changes in NAc α4*nAChR expression following MS suggesting this is most likely the case. However, it remains to be explored whether varenicline can attenuate alterations in plasma corticosterone and NAc α4*nAChR expression following chronic stress. If the length of exposure to stress impacts the efficacy of varenicline, similar to the length of ethanol and sucrose exposure, this will be an important consideration during the development of animal model for studying the effects of stress (particularly those which compare the effects of acute and chronic stress) and selection in future studies.

Another possible explanation for the lack of effect of varenicline on NAc α4*nAChR expression could be that changes in NAc α4*nAChR expression occur at a different time point to the one chosen for this study. We chose 20 h post treatment based on previous experiments conducted in our laboratory, using the acute administration of ethanol, cocaine and morphine, which produced maximum α4*nAChR expression changes in the ventral tegmental area (VTA, unpublished data). While we have explored α4*nAChR expression changes in the NAc at earlier time points (30 min, 1 h and 1.5 h) and found no detectable change in expression following treatment with varenicline and/or yohimbine (unpublished data), we have not examined any other times points. While 24 h post treatment appears to be the most common time point chosen for examining protein expression, changes in protein expression have been reported from 1.5 h to 88 days post-treatment [39,40,41,42,43]. Considering that these studies report changes in other brain regions following stress and few have conducted time-course experiments to determine the most appropriate time for protein expression analysis, it would be worthwhile exploring these aspects in future studies. Additionally, given the sex-specific effects of stress on the NAc found in this and other studies [6,44,45], it is also possible that varenicline could have a different efficacy for reducing yohimbine-stress effects in females. Further studies are necessary to confirm whether sex-specific roles for α4*nAChR exist during acute stress responses and how this might impact the efficacy of varenicline and the treatment of male and female patients in the clinical setting. 

In terms of chronic ELS, our modified MS model produced different effects in female and male mice. While both sexes displayed similar plasma corticosterone levels following exposure to MS and restraint, female control mice tended to be impacted by restraint more than female MS mice and males (increased body weight, significant increase in percent change in plasma corticosterone, and increased percent change in NAc α4*nAChR expression). Our findings both contrast and support Diehl and colleagues who found no difference in anxiety-like behaviours in MS male and female Wistar rats under basal conditions, but, following foot-shock stress, male MS rats demonstrated increased anxiety-like behaviour [46]. Similarly, Diehl and colleagues found male MS rats had reduced basal plasma corticosterone levels compared to controls. Following foot-shock there was no difference between controls and MS males. Interestingly, there was no difference between control and MS females under basal or stressful conditions in Diehl’s study. Whether the divergence in anxiety-like behaviours and plasma corticosterone response is related to species or methodological differences remains unclear. Nevertheless, it seems possible that the sex-specific effects of MS under basal and stressful conditions have different implications for the development of stress-related disorders in humans following exposure to traumatic childhood events. Certainly, human studies indicate that females are more likely to develop post-traumatic stress disorder following traumatic early life events compared to males [47]. The sex-specific effects of stress indicate that different mechanistic pathways are involved, and further studies are necessary to identify potential sex-specific therapeutic targets.

In this study, we demonstrated reduced NAc α4*nAChR expression under restrained conditions in MS males and no difference in percent change in NAc α4*nAChR expression following restraint stress in control and MS males and a reduced percent change in NAc α4*nAChR expression in MS females. This may indicate that exposure to MS renders the NAc less responsive to internal and external cholinergic signalling, potentially altering signalling in numerous pathways (for example, cholinergic, dopaminergic, and GABAergic) and subsequently the function of the NAc and other brain regions (such as the prefrontal cortex and hippocampus) with which it communicates. Alterations in cholinergic signalling have been implicated in the development of numerous psychiatric disorders, including alcohol use disorders, major depression, schizophrenia and PTSD. For example, in schizophrenia, it is hypothesised that cholinergic signalling into the NAc is inhibited and output to the PFC from the NAc is disinhibited (for review see [13]). Additionally, many of the disorders which are associated with altered cholinergic signalling are also more likely to occur in patients who have experienced traumatic events during childhood, and frequently with more severity [48,49,50,51,52,53]. Whether the changes in NAc α4*nAChR expression observed in this study result in behavioural changes which are associated with psychiatric disorders will need to be investigated in future studies. 

Given that ELS is also associated with an increased risk of developing AUDs later in life, it is noteworthy that studies have shown that MS increases ethanol consumption in male but not female rodents (for reviews, see [54,55]). Supporting this, human data indicate that males are more likely to engage in risky drinking behaviours compared to females [56]. Our findings showed MS males had greater change in plasma corticosterone levels in response to restraint stress. In this context, this could indicate the MS males either perceived the restraint as more stressful or were less able to cope with the effects of stress. Certainly, school children exposed to ELS display increased anxiety sensitivity compare to those who have not experienced ELS [57]. Furthermore, exposure to alcohol alters the ability to cope with stress, such that alcoholics attempting to remain abstinent have been reported to perceive stress more intensely than non-alcoholics [58,59,60,61,62]. Consistent with this, our laboratory has recently shown, in male mice, a single sedating dose of ethanol can increase α4*nAChR expression in presynaptic bouton of dopaminergic neurons projecting to the NAc [63]. While it is likely that the altered NAc α4*nAChR expression following restraint stress in MS males would alter dopaminergic signalling, contributing to increased ethanol consumption, our recent study and this study show that exposure to ethanol or stress alone can alter NAc α4*nAChR expression, potentially disrupting dopaminergic signalling and increasing susceptibility to substance use disorders. Furthermore, it is unknown how NAc α4*nAChR expression is impacted by ethanol consumption following exposure to MS. It would also be interesting to determine whether polymorphisms in the α4 subunit gene, CHRNA4, similar to those associated with increased risk for developing AUDs [64] and major depression disorder [65], alter perception of stress in males and females and whether this effect is amplified when polymorphisms in the glucocorticoid receptor and/or dopamine receptor genes co-exist with CHRNA4 polymorphisms. While the data presented in this study provide insight into the involvement of α4*nAChRs in stress, it is clear there are many more questions to be answered and more research is required to elucidate the mechanisms involved so that we can improve the life-time outcomes for those exposed to traumatic early life events.

## 5. Conclusions

Here, we have demonstrated for the first time that α4*nAChRs in the NAc are impacted by exposure to stress and that the systemic administration of varenicline can reduce plasma corticosterone levels in response to an acute stressor. The administration of varenicline has previously been shown to reduced sucrose, nicotine and alcohol consumption in humans and/or rodents [29,66,67,68]. While the evidence suggests varenicline modulates the release of dopamine in the NAc, dampening craving during abstinence [18]. Its ability to modulate stress has not previously been explored. Indeed, it would be highly beneficial for an anti-addiction therapeutic to dampen both stress and craving, especially considering the craving-intensifying effects of stress. While further studies are required to elucidate the mechanisms through which stress and nAChRs interact, varenicline may prove useful for alleviating the effects of stress, particularly during abstinence from substances impacting α4*nAChR-mediated pathways.

## Figures and Tables

**Figure 1 brainsci-08-00070-f001:**
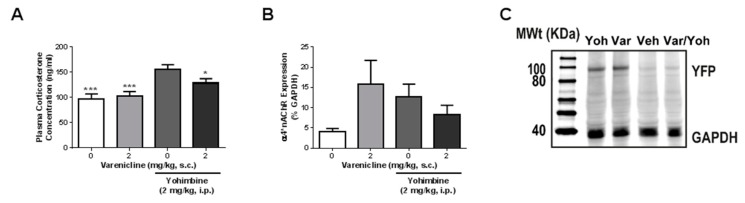
Varenicline attenuates yohimbine-induced increases in plasma corticosterone levels: (**A**) treatment with yohimbine caused a significant increase in plasma corticosterone levels compared to vehicle (*p* < 0.001), varenicline alone (*p* < 0.001) and varenicline with yohimbine (*p* < 0.05); (**B**) none of the varenicline/yohimbine treatment combinations had an effect on NAc α4*nAChR expression; and (**C**) representative Western blot image showing the effect of vehicle, varenicline alone, yohimbine alone and varenicline with yohimbine on tallow fluorescent protein (YFP) and Glyceraldehyde 3-phosphate dehydrogenase (GAPDH) expression. *n* = 5–12. One-way ANOVA with Bonferroni’s post hoc test. * *p* < 0.05, *** *p* < 0.001 compared to vehicle plus yohimbine. Veh: vehicle, Var: varenicline, Yoh: yohimbine.

**Figure 2 brainsci-08-00070-f002:**
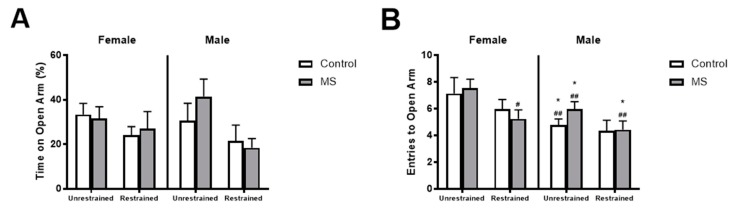
Effect of restraint stress on anxiety-like behaviours in female mice exposed to MS. In male and female mice, following exposure to maternal separation (MS, grey), 30 min of restraint stress had no effect on the (**A**) percentage of time spent on the open arm or (**C**) total distance travelled compared to control mice (white). In female mice, restraint reduced the (**B**) number of entries to the open arm (*p* = 0.485) in MS mice. Unrestrained female control and MS mice made more entries to the open arm than unrestrained male control and MS mice and restrained male MS mice. *n* = 7–11 per group. Three-way ANOVA with BKY’s post hoc test. * *p* < 0.05 compared to unrestrained female control mice. ^#^
*p* < 0.05, ^##^
*p* < 0.01 compared to unrestrained female MS mice.

**Figure 3 brainsci-08-00070-f003:**
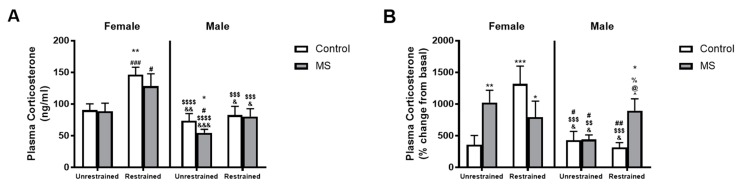
Sex-specific effects of MS and restraint stress on plasma corticosterone levels. Exposure to restraint increased plasma corticosterone levels compared to unrestrained mice in female control (white) and MS mice but had no effect in male mice (**A**). However, expressing plasma corticosterone levels as the percentage change from basal levels showed a significantly greater plasma corticosterone rise in restrained male MS mice compared to all other male groups (**B**). In females, unrestrained controls displayed a small change from basal plasma corticosterone levels than all other female groups. *n* = 6–19 per group. Three-way ANOVA with BKY’s post hoc test. * *p* < 0.05, ** *p* < 0.01 and *** *p* < 0.001 compared to unrestrained female control mice. ^#^
*p* < 0.05, ^##^
*p* < 0.01, and ^###^
*p* < 0.001 compared to unrestrained female MS mice. ^$$^
*p* < 0.01, ^$$$^
*p* < 0.001, and ^$$$^^$^
*p* < 0.0001 compared to restrained female control mice. ^&^
*p* < 0.05, ^&&^
*p* < 0.01 and ^&&&^
*p* < 0.001 compared to restrained female MS mice. ^%^
*p* < 0.05 compared to unrestrained male control mice. ^@^
*p* < 0.05 compared to unrestrained male MS mice. ^^^
*p* < 0.05 compared to restrained male control mice.

**Figure 4 brainsci-08-00070-f004:**
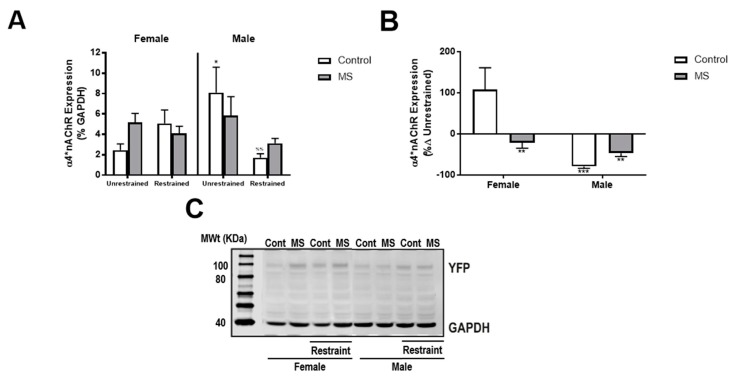
MS attenuates restraint-induced reductions in NAc α4*nAChR expression in male mice. (**A**) In male control (white) but not MS (grey) mice, exposure to restraint caused a decrease in NAc α4*nAChR expression. No effects were identified in female mice. (**B**) The percentage change in NAc α4*nAChR expression from unrestrained conditions was greatest in female control mice. (**C**) Representative Western blot image showing the effect of sex, MS and restraint on YFP and GAPDH expression. *n* = 4–11 per group. Three-way ANOVA with BKY’s post hoc test. * *p* < 0.05 compared to female unrestrained control mice. ^%%^
*p* < 0.01 compared to unrestrained male control mice. ** *p* < 0.01, *** *p* < 0.001 compared to female control mice.

**Table 1 brainsci-08-00070-t001:** Sex-specific effects of maternal separation on body weight.

Sex	Control	MS
Female	17.01 ± 0.333	17.97 ± 0.294 *
Male	21.64 ± 0.446	22.17 ± 0.277

* *p* < 0.05 compared to female controls. MS: maternal separation

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
