# Peer review of "Sex Specific Alterations in α4*Nicotinic Receptor Expression in the Nucleus Accumbens"

_brainsci, 2018, doi:10.3390/brainsci8040070_

Round 1

Reviewer 1 Report

The authors report that in mice the effects of various stresses on AChR alpha 4 subunits are sex-dependent.  Some stresses altered alpha 4 amounts, while others did not. They found that varenicline attenuated yohimbine-induced increases in corticosterone. Varenicline can reduce consumption of nicotine and alcohol.  They suggest that it might also prove useful for reducing stress.  Additional studies of varenicline effects on the stresses they studied would have been useful.

They point out that childhood stress increases the probability of smoking, alcohol use disorders, depression and other self-destructive behaviors.  Stress, addiction, and many mental diseases are associated with alterations of the nucleus accumbent and alterations in dopaminergic signaling which are mediated by alpha 4 AChRs, as is glucocorticoid release. Others have reported that mecamylamine, a non-competitive AChR antagonist, blocks stress-induced dopamine release in the nucleus accumbens.  Varenicline is a high affinity low efficacy agonist agonist that is very effective at desensitizing alpha 4 AChRs, a net antagonist effect.

Their study is very well written, for the most part.

The mouse line with fluorescent labeled alpha 4 subunits was developed in Henry Lester's lab.  The labeled subunits provided an elegant and convenient method for measuring AChR amounts in response to stress and varenicline.  A range of stresses and behavioral assays provided substantial breadth to their studies.

Their data are clearly presented, and their conclusions are generally reasonable.

In their abstract they conclude that "Varenicline may prove useful for alleviating the effects of stress."  But in the discussion the say "However, it remains to be explored whether varenicline can block the effects of chronic stress, systematically and/or centrally."  Which is it?  Or in the latter case is the reference only to the amount of alpha 4 expression or to chronic versus acute yohimbine stress?  This is not as clear as it should be.

Author Response

We have reworded the last sentence of the abstract to state that while varenicline attenuated acute stress-induced rises in corticosterone levels, future studies are required to determine whether varenicline is effective for relieving the effects of stress.

Reviewer 2 Report

The current study by Holgate and Bartlett reports on the acute effect of varenicline on stress stimulated corticosterone secretion and expression of accumbal alpha4 containing nicotinic acetylcholine receptors as well as the impact of early life stress on these parameters, and body weight chane and anxiety-like behaviors in male and female mice. The results showed that varenicline modestly reduced yohimbine-induced increase in corticosterone secretion but increased expression of alpha4 containing receptors in the nucleus accumbens. Maternal separation increased body weight in female but not male mice. Anxiety-like behaviors were not changed in female mice exposed to maternal separation but were increased in male mice. Exposure to early life stress reduced basal plasma but did not alter restrained stress induced stimulation of corticosterone secretion in males or females compared to their controls. Yet, based on percent changed compared to ba saline values, there appear to be changes in a sex related manner. Expression of alpha4 containing receptors is also follows a similar pattern. Overall, the manuscript is well written and the data are of interest to the readers of the journal. However, there are some alternative interpretation of the data as well as some needs for providing further details about the experimental design and possibly alternate analyses which are described below:

Major

The title and emphasis of the paper is on sex differences but the male and female mice were not compared directly. Only inference was made in each sex based on the control mice of the same sex.

the raw baseline data should analyzed with restrained to show how effective stress was in each group and three way ANOVA should used to simultaneously analyze the effect of early life stress, the sex as well as  impact of restraint stress on plasma corticosterone. The same should be considered for EPM data.

Animals were anesthetized and blood samples collected for the measurement of stress hormones. Aren't these procedures stressful?

I am a bit concerned about huge variability in  baseline values in different experiments. Is that the impact of double injection or something else?

Are the controls were weaned at the same age and housed individually?Isolation and handling daily appear to caused behavioral and hormonal changes.

the section on the effect of varenicline on yohimbine-induced changes in plasma cort can be part of another study and thus can deleted.

Minor

1. Please change the title if there is no sex related difference in the study when the data is re analyzed

2. Line 40, please change "lead" to "leading".

3. Line 48, please change "preventing lifelong" to "preventing its lifelong".

4. Line 82, please delete the extra space.

5. Line 83, please chane was to were.

6. First sentence of section 3.2 appears to belong to the previous section. Please verify.

7. Line 217, please change determined to determining.

8. Please show the body weight data as a table.

9. Please modify the heading of section 3.3 since additional analyses may result in a different outcome.

10. Please change basal to no restrained since basal can be confusing as if the mice were tested for basal EPM as well as after restraint stress.

11. Although time on the open arm is lower between basal (non-restrained) and restraint male mice, there is no difference between restrained control vs. MS restraint groups.

12. Line 217, please change "there no" to "there was no".

13. Line 362, please change "be impacted" to "to be impacted" .

14. Line 365, please change to Diehl and colleagues'' study.

15. Line 372, nevertheless.

16. Please delete the first sentence of the Acknowledgment section. It is repeated below under COI.

Author Response

Major

1.     The title and emphasis of the paper is on sex differences but the male and female mice were not compared directly. Only inference was made in each sex based on the control mice of the same sex.

the raw baseline data should analyzed with restrained to show how effective stress was in each group and three way ANOVA should used to simultaneously analyze the effect of early life stress, the sex as well as  impact of restraint stress on plasma corticosterone. The same should be considered for EPM data.

The data has been reanalysed using 3-way ANOVA to enable direct comparison between males and females.

2.     Animals were anesthetized and blood samples collected for the measurement of stress hormones. Aren't these procedures stressful?

Many experimental manipulations (including moving the cage, removing a cage mate,) and anaesthetics alter corticosterone levels. It is not possible to collect blood without altering plasma corticosterone as the procedure itself is stressful. However, we treated all test and control in mice the same way during the experimental procedures. All mice were tested within the same time period and provided with at least 1 hour to habituate to the test room before any experimental procedure commenced. We also used a pseudorandom Latin square design for deciding the order mice from each group underwent experimental procedures (such that mice were tested in batches and each batch contained one mouse from each group and the order the mice were tested was different for each batch). We have modified the methods section to improve clarity regarding the controls used in this manuscript.

3.     I am a bit concerned about huge variability in baseline values in different experiments. Is that the impact of double injection or something else?

This is not unexpected as administration of injections produce a stress response.The corticosterone levels following the double injection of vehicle are elevated compared to unrestrained mice but lower than those obtained using restraint alone. This suggests the injection procedure itself is stressful, although not as stressful as restraint or yohimbine. However, given that yohimbine and restraint produced similar rises in corticosterone, the possibility that yohimbine may not impact the same brain pathways as restraint and the lack of significant changes in NAC a4YFP expression in the acute study, we opted for the more traditional and conservative method of restraint for the chronic studies.

4.     Are the controls were weaned at the same age and housed individually? Isolation and handling daily appear to caused behavioral and hormonal changes.

The controls were weaned at the same age as the MS and housed individually and handled exactly as the MS mice were. The only difference between the two groups is the control mice and mother were placed in the same novel cage during the MS conditioning phase, whereas the MS pups were placed in a separate novel cage to the mother. We have modified the methods section to make this aspect clearer.

5.     The section on the effect of varenicline on yohimbine-induced changes in plasma cort can be part of another study and thus can deleted.

While we could present these findings as two separate papers, we feel these results provide greater insight into the effects of stress on NAc a4*nAChRs and the role they may play in stress-related disorders when considered together. Both the acute and chronic studies indicate that NAc a4*nAChRs are involved in acute and chronic stress responses and provide information which may help elucidate why traumatic stress leads to other disorders like addiction. Together these result support our previous findings that the length of exposure to ethanol or sucrose is critical for inducing changes in brain morphology and function. Additionally, the length of exposure to alcohol and sucrose impacted the efficacy of varenicline, and other nAChR modulators. As this appears to also be the case for stress, this will need to be considered in future studies aimed at elucidating underlying mechanisms and during the development of animal models for identifying potential therapeutic interventions for treating stress-related disorders. We have modified the discussion to make the combined implications of acute and chronic studies clearer.

Minor

1.     Please change the title if there is no sex related difference in the study when the data is re analysed.

As sex related differences were identified, we have not changed the title.

2. Line 40, please change "lead" to "leading".

The correction has been made to the text.

3. Line 48, please change "preventing lifelong" to "preventing its lifelong".

The correction has been made to the text.

4. Line 82, please delete the extra space.

The correction has been made to the text.

5. Line 83, please change was to were.

The correction has been made to the text.

6. First sentence of section 3.2 appears to belong to the previous section. Please verify.

The correction has been made to the text.

7. Line 217, please change determined to determining.

The correction has been made to the text.

8. Please show the body weight data as a table.

We have replaced the graph with a table.

9. Please modify the heading of section 3.3 since additional analyses may result in a different outcome.

The heading has been modified.

10. Please change basal to no restrained since basal can be confusing as if the mice were tested for basal EPM as well as after restraint stress.

Basal has been changed to unrestrained, except where basal blood samples were collected.

11. Although time on the open arm is lower between basal (non-restrained) and restraint male mice, there is no difference between restrained control vs. MS restraint groups.

Following reanalysis using 3-way ANOVA we found no effects on time on the open arm for sex, restraint or MS. The text has been adjusted.

12. Line 217, please change "there no" to "there was no".

The correction has been made to the text.

13. Line 362, please change "be impacted" to "to be impacted" .

The correction has been made to the text.

14. Line 365, please change to Diehl and colleagues'' study.

The correction has been made to the text.

15. Line 372, nevertheless.

The correction has been made to the text.

16. Please delete the first sentence of the Acknowledgment section. It is repeated below under COI.

The correction has been made to the text.

Round 2

Reviewer 2 Report

Minor

Line 166, something is missing in this sentence. May be it should be:

.........incubated in equal volumes with 1:100 steroid.....

Lines 193 - 194

2-way ANOVA was used to compare the effects of maternal separation and sex on the percentage ......

Line 264, please delete "no interaction of" since it is repeated in the next line.

Line 268, please change "and interaction" to "an interaction".

Line 408, during animal model development is not clear. Please state which animal model.

Author Response

Line 166, something is missing in this sentence. May be it should be:

.........incubated in equal volumes with 1:100 steroid.....

The sentence has been corrected.

Lines 193 - 194

2-way ANOVA was used to compare the effects of maternal separation and
sex on the percentage ......

The sentence has been corrected.

Line 264, please delete "no interaction of" since it is repeated in the
next line.

The correction has been made.

Line 268, please change "and interaction" to "an interaction".

The correction has been made. 

Line 408, during animal model development is not clear. Please state

which animal model.

We have corrected the sentence to clarify that we are referring to animal models for studying stress and comparing the effects of acute and chronic stress.